# Remote Learning Experience and Adolescents’ Well-Being during the COVID-19 Pandemic: What Does the Future Hold?

**DOI:** 10.3390/children9091346

**Published:** 2022-09-02

**Authors:** Riki Tesler

**Affiliations:** 1Department of Health Systems Management, School of Health Sciences, Ariel University, Ariel 4077625, Israel; rikite@ariel.ac.il; Tel.: +972-454-300-7323; 2Health Promotion and Well-Being Research Center, Department of Health Systems Management, School of Health Sciences, Ariel University, Ariel 4077625, Israel

**Keywords:** adolescents, COVID-19, remote learning, well-being, self-rated health, psychosomatic symptoms

## Abstract

**Background**. Major shifts within the education system have taken place during the COVID-19 pandemic; frontal teaching was often replaced with remote learning, which has affected students in many ways. We investigated the associations and predictors of perceptions of the remote learning experience on well-being (life satisfaction, self-rated health, psychosomatic, and psychological symptoms). **Methods**. We conducted a cross-sectional research study consisting of 1019 school students in Israel aged 11–18 (53.5% girls, 46.7% boys). Questionnaires were distributed from May–July 2021 during school time. The percentages of participants with various levels of well-being (WB) and remote learning experience were compared. Multiple regression procedures were used to analyze factors predicting wellbeing. **Results**. All of the remote learning items had statistically significant positive correlations with life satisfaction and self-rated health (i.e., better overall WB was associated with a more positive perception of the remote learning experience). Male gender, high socioeconomic status, greater involvement in lessons in the past year, and connection to the pedagogical team/school and peers predicted better overall WB (F-ratio = 14.03; *p* < 0.01; adjusted R^2^ = 0.08). **Conclusions**. Our results highlight the need for schools to target youths’ coping skills, which may lead to better remote learning experiences. These findings also provide several implications for the need to support children and adolescents through positive activities, relaxation/mindfulness, and cognitive coping to deal with the psychosomatic symptoms during remote learning periods.

## 1. Introduction

COVID-19 was declared as a global pandemic by the World Health Organization in March 2020 [1]. Restrictions were enacted around the world, most of which caused disruption to the daily life of children and adults alike, many affecting education systems [2]. 

For children and adolescents, this has perhaps been most notable in the area of education. In March 2020, nearly 1.87 million Israeli youth from kindergarten through 12th grade transitioned from in-person education to remote learning platforms [3]. Previous studies have identified this phenomenon as emergency remote teaching, describing a temporary change of instruction as a result of a crisis or traumatic situation [4,5]. Emergency remote teaching has many technological, pedagogical, and social challenges for both teachers and students [6,7,8]. According to a report by the Organization for Economic Cooperation and Development, children’s daily lives have been impacted by the COVID-19 pandemic. School and leisure activities for children have been affected, many activities being offered online instead of in-person. Children were expected to spend more time on digital devices in order to stay in school and connected with others [9].

Several theoretical articles have postulated the impact of remote learning on both typical and vulnerable populations regarding social difficulties, compromised learning, and negative impacts on parent–child relationships [8,10]. However, limited empirical work has explored the impact of the COVID-19 pandemic on remote learning experiences [11]. Thus, investigating the impact of remote learning on students’ well-being holds exceptional psychological, educational, and social importance.

### Risk and Protective Factors for Remote Learning

Although the change from in-person instruction to remote instruction has affected all students, Reich et al. (2020) point out the phenomenon of six separate educational disparities during the COVID-19 pandemic [12]. 

Age: In terms of technical skill development, many young people lack the self-confidence to use a digital platform for learning or have not yet developed the skills required to use technology in deep and critical ways, especially younger children [12,13]. 

Socioeconomic Background: Students from higher socioeconomic status (SES) backgrounds are more likely to have acquired the necessary critical thinking skills associated with selectively accessing and assessing technology content, as well as having the financial means to gain help [13,14,15]. Moreover, studies indicated that financially disadvantaged groups were less likely to engage in remote learning, and thus students from low-income families naturally struggled more with the sudden shift to remote learning [16,17].

Grade: A study conducted in Poland on the difficulties of remote learning as perceived by students during the coronavirus pandemic found that the oldest students (17–18-year-olds) and those living in rural areas reported more difficulties related to learning, mainly due to teachers’ higher requirements and poor organization, in addition to technical difficulties and insufficient skills in using software [18,19]. 

Gender: Previous research has shown that there have been gender differences in coping with remote learning, influencing academic outcomes in response to digital learning. Stein-Zamir (2020) found that boys may have an advantage over girls regarding online learning due to a higher perceived ability and engagement with computers [3]. However, others have suggested that girls have more negative beliefs about learning via computer and digitally than boys [5,6]. Yet the issue is still under debate and these studies are not conclusive; others have suggested that there are no differences between male and female students’ attitudes regarding digital learning [8].

Historical Performance: Specifically, as suggested by Reich et al. (2020), students who were doing well academically were likely to continue doing well remotely [12], whereas students struggling academically were likely to have a harder time engaging with academic materials remotely, often with limited support from a teacher [19,20]. 

Adaptation and Regulation Skills: children with better adaptive coping and emotion regulation skills may be better equipped to navigate the challenges of remote learning [14,16].

As should be clear from this list, there are many possible disadvantages in the remote learning environment, and every child’s experience is different. Regardless of the initial cause, any of the factors above can trigger psychological distress such as anxiety and depression, which may increase as students start to lose school connectedness, including a loss or lessening of the belief that both adults and peers in their school care about their learning as well as about them as an individual [14,21,22,23]. In response, the obvious remedy is emotional support for both learners and their families during crises. This effort is essential, and support should be outreaching and proactive to ensure that the families that are most impacted by this situation are managing emotionally, financially, and logistically [24,25,26]. 

Though much research has focused on remote learning among children and adolescents, few studies have described the underlying well-being among this population that impacts (positively and negatively) the process of remote learning. Additionally, this is the first study to our knowledge that focuses on the association of remote learning with sociodemographic characteristics and well-being among children and adolescents in an Israeli setting.

The two primary aims of this study were: (a) to examine children and adolescents’ remote learning experience during COVID-19, and (b) to explore associations between the remote learning experience, sociodemographic characteristics, and well-being and to examine what factors predict well-being.

## 2. Materials and Methods

### 2.1. Research Design 

A cross-sectional study design was implemented. The study consisted of 1019 Israeli students aged 11–18. Questionnaires were distributed to the students between May and July 2021 during school time.

### 2.2. Participants

The target population of the study was students in grades 5–12 in Israeli schools. Students were given the option to participate or not. The sampling method was probabilistic random sampling within layers. Participating students were asked to fill out a questionnaire during one of their classes during remote sessions. The classes were sampled from the most updated list from the Israeli Ministry of Education by age group, geographical distribution by sector, and type of school. The sample included a total of 1019 Israeli students aged 11 to 18 (53.3% girls, 46.7% boys; chi-squared = 21.47; *p* < 0.001). The representation of students from 12th grade was the lowest (*n* = 56, 5.5% of the sample; chi-squared = 78.96; *p* < 0.001; Table 1).

### 2.3. Data Collection

The average response rate was 25.5%. The research tool was an online anonymous self-completion questionnaire. The questionnaire includes the HBSC core questionnaire and new questions on the effects of COVID-19. The HBSC is a large cross-national study that has been used to examine the health and lifestyle determinants of school age children in 52 countries and regions for over 30 years. The methodology was consistent with that of the International HBSC Survey. The sampling error was ±3.1% at a 95% confidence level [27].

This study received approval from the Israeli Ministry of Education, as well as the Ethics Committee of Israel’s Chief Scientist Office.

### 2.4. Independent Variables 

The HBSC includes items describing participants’ sociodemographic characteristics [27,28]. More specifically, study participants reported their self-identified gender (male or female), grade (5th through 12th), and change in socioeconomic status. Change in socioeconomic status was evaluated using the question: “according to your opinion, in comparison to before corona virus 2019 pandemic, in the past 30 days, what is the socioeconomic status of the family.” The question refers to the last 30 days in order to learn about the average socioeconomic status. Answers consisted of the following options: “much worse”, “less good”, “no change”, “better”, and “much better”. Answers were combined to create the following scores: 1 point—“much worse or less good, 2 points—“no change”, and 3 points—“better or much better”. 

### 2.5. Remote Learning Experience

Two broad categories of remote learning experience were evaluated: attitudes toward remote learning, and remote learning and social connectedness.

Attitudes toward remote learning consisted of the following questions: (1) In the past 12 months, how much were you involved in lessons conducted remotely (via Zoom). Answer options were: “I did not attend lessons”, “I connected with the camera off and did not listen to the lesson”, “I connected to the lesson with camera off and listened”, “I connected to the lesson with camera on and listened without being active”, and “I connected to lesson with camera on and was active”. Answers were coded in the following way: “I did not attend lessons” or “I connected with camera off and did not listen to the lesson” (1 point); “I connected to the lesson with camera off and listened” (2 points); “I connected to the lesson with camera on and listened without being active” (3 points); and “I connected to lesson with camera on and was active” (4 points). The second question was: “How do you feel about the lessons being conducted remotely?” Answers included: “dislike very much” (1 point); “dislike slightly” (2 points), “like slightly” (3 points); “like very much” (4 points).

### 2.6. Remote Learning and Social Connectedness 

Remote learning and social connectedness was assessed via the following two questions: (1) “During COVID-19, while studying remotely, how much did you feel connected or disconnected from the pedagogical team and school?” and (2) “During COVID-19, while studying remotely, how much did you feel connected or disconnected from your peers?” For both questions, the following scale was used: “very much disconnected” (1 point); “slightly disconnected” (2 points); “slightly connected” (3 points); “very connected” (4 points); and “very much connected” (5 points) [28]. 

### 2.7. Dependent Variables

The dependent variables consisted of well-being outcomes (life satisfaction, self-rated health, and psychosomatic and psychological symptoms). All measures were adopted from the HBSC survey mandatory questions protocol.

Life satisfaction was measured by the Cantril’s ladder [27], an efficient, global measure with high construct validity [28]. Young people were presented a picture of a ladder with steps ranging from 0 to 10 (where 10 represented the best possible life and 0 the worst). They were asked to indicate where on the ladder they would place their life at present. This scale was used as a continuous measure. 

### 2.8. Self-Rated Health 

SRH is a standardized indicator that has been used extensively in various health research. For the current analysis, self-rated health was assessed using the question: “How would you describe your health in the past year?” Answers were scored on a 4-point scale ranging from 1 (excellent) to 4 (poor) and were then recoded to 1 (excellent), 2 (good), and 3 (poor and fair) [27].

### 2.9. Psychosomatic and Psychological Symptoms

Psychosomatic and psychological symptoms were assessed using eight questions pertaining to pain (headache, abdominal, back), feelings (bad mood, anger, irritability), sleeping (difficulties sleeping or falling asleep) and dizziness. Each question was scored on a 5-point scale ranging from “almost every day” (1) to “rarely or never” (5). A total score was computed. In accordance with the HBSC methodology protocol, two independent indexes were created: (1) somatic complaints (abdominal pain, headache, back pain, dizziness); and (2) psychological complaints (nervousness, depression, bad mood/irritability, difficulty sleeping). In both indexes, lower scores represent more psychosomatic complaints [27,28].

### 2.10. Data Analysis

Demographic characteristics, well-being, and attitudes toward remote learning are described using descriptive statistics (*n*, percentage, and median). This analysis was conducted for the entire group based on the sociodemographic characteristics.

None of the dependent variables met the normal distribution assumption. Therefore, a non-parametric statistic was used. More specifically, sociodemographic differences in prevalence of the three self-rated health categories (poor or fair, good, excellent) were compared using the chi-squared test. Differences in sociodemographic characteristics in the other two well-being measures (life satisfaction, and psychosomatic and psychological symptoms) were compared using the Mann–Whitney U test for dichotomized variables (gender) and the Kruskal–Wallis test for multilevel categorical variables (grade and change in socioeconomic status). For the psychosomatic and psychological symptoms variable, differences between females and males were presented using a box-plot figure. In the figure, the central box is representative of the values from the lower quartile to the upper quartile. The vertical line extends from the minimum value to the maximum value, while excluding the outside values (as they are displayed as separate points). An outside value was defined as either: (1) a value lower than the lower quartile minus 1.5 times the interquartile range, or (2) a value higher than the upper quartile plus 1.5 times the interquartile range. The middle line is representative of the median. 

Associations between participants’ attitudes toward the remote learning experience and sociodemographic characteristics and well-being measures were examined using Spearman’s rank correlations. Subsequently, four separate multiple regression procedures to analyze factors predicting each of the well-being measures were conducted. All independent variables were checked for multicollinearity by using the variance of inflation factor (variance of inflation factor > 10). For continuous variables, only variables that were statistically significantly correlated with the well-being measures were entered into the model. Similarly, for categorical variables, only variables that statistically significantly differed between participants with different well-being measures were entered into the model. 

For all statistical analyses, SPSS Statistics for Windows, version 23 (SPSS Inc., Chicago, IL, USA), was used; the level of significance was set to *p* < 0.05 (2-tailed).

## 3. Ethical Considerations

The study protocol was approved by the Ethics Committee of Ariel University, confirmation number: RO10,203; 21 May 2021. A preliminary letter regarding the survey was sent to the parents of the students. They were asked to confirm their children’s participation. On the day of the survey, it was made clear to the students that the questionnaire was anonymous, and that their names should not be written on it.

## 4. Results

### 4.1. Sociodemographic Characteristics of the Study Participants 

The total number of participants was 1019 students in the 5th (*n* = 121, 11.9% of the sample) to 11th and 12th grades (*n* = 177, 17.3% of the sample). Most study participants were female (*n* = 544, 53.3% of the sample) and Jewish (*n* = 795, 78.0% of the sample; Table 1). 

Participants’ well-being measures according to sociodemographic characteristics.

Regarding self-rated health, statistically significant differences were found in all sociodemographic characteristics (Table 2). More specifically, among males, the percentage of participants reporting “poor and fair” overall health was greater than in females (52.31% of males vs. 42.09% of females; Chi-square = 10.61; *p* = 0.01). Compared to the 5th grade, the percentages of participants with “good” health in the other grades were statistically significantly lower (chi-squared = 9.69, *p* = 0.001). The percentage of participants with “excellent” health belonging to the “better/much better” and “the same” socioeconomic status (51.90 and 49.09%, respectively) was statistically significantly greater than the percentage observed in “the same” socioeconomic status group (35.12%; chi-squared = 12.08; *p* = 0.0005). 

In contrast to self-rated health, females presented lower scores for somatic and psychological complaints compared to males (i.e., more complaints; *p* < 0.0001; Figure 1). Similar to self-rated health, in comparison to older students and to students who reported worse socioeconomic status, younger students and students who reported better socioeconomic status presented fewer somatic and psychological complaints, as well as greater life satisfaction (*p* range: <0.0001 to 0.002; Table 3).


Participants’ remote learning experience.


When looking at the entire group, 18.4% (*n* = 188) and 41.9% (*n* = 427) of the sample did not attend lessons or connected and listened to the lesson with the camera off, respectively. Moreover, 13.0% (*n* = 132) and 30.1% (*n* = 307) of the sample very much liked remote learning or disliked it very much, respectively. Regarding remote learning and social connectedness, 9.32% (*n* = 95) of the sample felt very much connected to the pedagogical team in comparison to 12.65% (*n* = 129) who felt very much disconnected. In the connection to peers, percentages of students who felt very much socially connected was slightly higher (*n* = 291, 28.55%; Table 4).

Regarding gender differences, in comparison to females, a greater percentage of males very much liked remote learning (males: 15.96%, *n* = 76; females: 10.29%, *n* = 56; chi-square = 11.98, *p* < 0.001). The percentage of females who felt slightly disconnected from peers (*n* = 76, 13.97%) was greater than that of males (*n* = 40, 8.40%; chi-square = 13.69, *p* < 0.001).

### 4.2. Association between Participants’ Remote Learning Experience and Sociodemographic Characteristics 

In all remote learning items, statically significant correlations were found with grade (i.e., the higher the grade, the more negative the attitudes toward remote learning, except for connectedness with peers; Table 4). 

### 4.3. Association between Participants’ Remote Learning Experience and Well-Being Measures

In all remote learning items, statistically significant correlations were found with well-being measures, i.e., better well-being was associated with better remote learning experience (r ranges from 0.06 to 0.28: *p* = 0.0001; Table 5).

### 4.4. Prediction of Well-Being

According to the multiple regression analysis conducted, both sociodemographic characteristics and remote learning experience predicted all the well-being outcomes. More specifically, in the sociodemographic variables, being a female and reporting an adverse change in socioeconomic status predicted less favorable well-being outcomes in three indexes. In the remote learning variables, greater involvement in lessons and better connectedness with peers predicted better well-being in all four indexes assessed. Lower connectedness to the pedagogical team predicted lower well-being in three indexes. The four models’ adjusted R^2^ ranged from 0.07 (psychosomatic symptoms) to 0.14 (psychological symptoms and life satisfaction) with *p* < 0.0001. For additional information, refer to Table 6. 

## 5. Discussion

The COVID-19 pandemic is revolutionizing education globally, and not only in Israel [7,8,9]. Schools have been forced to use technology and media to continue their educational activities [10].

This study was the first to examine remote learning experiences during COVID-19 among a sample of Israeli students. We found that over 50% of the sample disliked remote learning, and more than a third reported that they did not actively participate in the remote lessons. 

Male students had slightly more favorable attitudes toward remote learning, with higher percentages reporting that they “like very much” remote learning compared to female students. Female students tended to participate with their camera off compared to male students. These gender differences are consistent with other studies that have reported that males have overall more favorable attitudes toward technology when compared to women [29,30]. However, a lower prevalence of females participating with the camera off may not be directly related to negative attitudes toward remote learning. It may be related to other factors, such as, for example, possible concerns regarding their appearance on camera. Therefore, gender differences in camera usage should be further explored.

Surprisingly, we found that children in lower grades had more favorable attitudes toward remote learning compared to older students. This finding somewhat contradicts other studies reporting greater difficulties with remote learning among younger children [14,31]. 

However, such studies examined how parents perceived the remote learning of their children [10,32], whereas our study investigated the children’s perceptions and attitudes. In addition, children attending elementary schools in Israel had overall more in-person learning opportunities compared to those in higher grades, who continued to learn remotely when the younger children had already returned to school [26], which may explain why their overall remote learning experience was more negative. 

Our second objective was to examine factors predicting adolescents’ well-being. The present study found that overall, remote learning experience predicted well-being. More specifically, a higher level of involvement in remote learning and greater connectedness to school pedagogical teams and friends predicted greater life satisfaction, better overall health, and fewer psychosomatic symptoms. Peer support and teacher support have been often linked to better health outcomes among adolescents [33,34]. It is possible that active participation in lessons also increased the level of connectedness with both the school and schoolmates, which, in turn, was associated with the level of support from peers and teachers.

While school grade, a change in SES, and feelings toward remote learning were associated with the well-being measures, they did not emerge as significant predictors. Younger students are generally more protected by their parents, which could be the reason why their answers regarding the SES change questions were neutral or more positive compared to those given by the older students. A decline in SES could reflect job or business loss experienced by one or both parents. Recent studies conducted during the pandemic have found that parental stress associated with SES worries could affect the child’s well-being [35].

It is also important to note the gender differences in well-being. Apart from self-rated health, in which female students reported better overall health, they scored significantly lower on all other well-being measures (life satisfaction, and psychological and somatic symptoms) compared to male students. This finding is consistent with other studies showing that females consistently tend to report more mental health problems than males [27,36,37,38] The males’ poorer self-rated health could reflect the specific conditions of the pandemic in which social distancing restrictions have lessened their physical activity levels and increased their sedentary behavior, all which are associated with poorer health [39,40,41,42].

### Strengths, Limitations, and Directions for Future Study 

Our study had several strengths. This was the first study in Israel, with findings that can be used to generate suggestions within the field of education and online well-being. It sheds light on students’ attitudes and emotional reactions to remote learning during a time of crisis. The present study also has a few limitations. First, the study used a cross-sectional design, which does not allow inferences on causal relationships. A future longitudinal design may be able to better reflect the direction of the associations established in the present study. Second, the study variables were all self-report measures, which may be subject to common method biases [43]. In addition, due to the pandemic, some of the data were collected while the children were attending remote zoom lessons, without the active presence of the teacher or research assistant. This may have impaired participation rates and skewed our results toward more positive outcomes. Another limitation was that we did not include additional variables in the study (e.g., nutrition levels, accommodation), which could have enriched the findings, specifically for low SES participants.

## 6. Conclusions

The present study demonstrated that negative attitudes and experiences with remote learning, along with decreasing connectedness with the school framework and classmates, may have severe consequences for adolescents’ well-being. The results of this study highlight the need for a better education policy, and interventions that emphasize student connectedness and engagement with the school educational teams, as well as with peer students in times of crises, when there is an immediate need for a transition to remote learning. It is imperative that coping strategies are taught to children in order to help them overcome any issues or barriers associated with the COVID-19 pandemic. In addition, special attention should be given to girls, to students in higher grades, and to those whose families suffered additional trauma (such as loss of income).

## Figures and Tables

**Figure 1 children-09-01346-f001:**
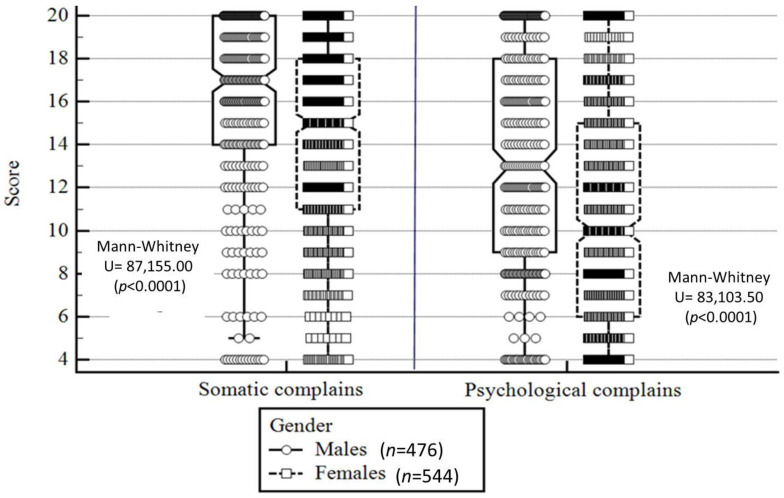
Somatic and psychological symptoms—gender differences. Notes: Higher score represents fewer somatic and psychological symptoms; the central box represents the values from the lower to upper quartile (25–75 percentiles); the vertical line extends from the minimum to the maximum value, excluding outside values which are displayed as separate points. An outside value is defined as a value that is smaller than the lower quartile minus 1.5 times the interquartile range, or larger than the upper quartile plus 1.5 times the interquartile range; the middle line represents the median.

**Table 1 children-09-01346-t001:** Sociodemographic characteristics of study participants (*n* = 1019).

Variable		*n* (%)
Grade	5th	121 (11.9)
	6th	118 (11.5)
	7th	163 (16.0)
	8th	148 (14.5)
	9th	159 (15.6)
	10th	134 (13.1)
	11th + 12th	177 (17.3)
Gender	Female	544 (53.3)
	Male	476 (46.7)
Change in Socioeconomic status, score	Much worse/worse	205 (20.1)
	The same	605 (59.3)
	Better/much better	210 (20.6)

**Table 2 children-09-01346-t002:** Participants’ self-rated health according to sociodemographic characteristics.

Variables		Self-Rated Health		
		Poor or Fair: *n* (%)	Good: *n* (%)	Excellent: *n* (%)
Total group (*n* = 1019)		217 (21.3)	323 (31.7)	479 (46.9)
Gender	Male (*n* = 476)	249 (52.31)	135 (28.36)	91 (19.11)
	Female (*n* = 544)	229 (42.09)	188 (34.55)	126 (23.16)
	*Chi-square* (*p* value)	**10.61 (0.001)**	**4.49 (0.03)**	2.48 (0.11)
Grade	5th (*n* = 121)	22 (18.18)	50 (41.32) ^b–g^	49 (40.49) ^b^
	6th (*n* = 118)	21 (17.79)	33 (27.96) ^a^	64 (54.23) ^a^
	7th (*n* = 163)	32 (19.63)	50 (30.67) ^a^	80 (49.07) ^f^
	8th (*n* = 148)	30 (20.27)	45 (30.40) ^a^	73 (49.32) ^f^
	9th (*n* = 159)	36 (22.64)	49 (30.81) ^a^	73 (45.91)
	10th (*n* = 134)	31 (23.13)	52 (38.80) ^a,g^	50 (37.31) ^c,d,g^
	11th + 12th (*n* = 177)	44 (24.85)	43 (24.29) ^a,f^	89 (50.28) ^f^
	*Chi-square* (*p* value)	2.04 (0.15)	**9.69 (0.001)**	**10.69 (0.001)**
Change in Socioeconomic status	Much worse/worse (*n* = 205)	60 (29.26)	72 (35.12)	72 (35.12)
	The same (*n* = 605)	125 (20.66) *	183 (30.24)	297 (49.09) *
	Better/much better (*n* = 210)	33 (15.71) *	68 (32.38)	109 (51.90) *
	*Chi-square* (*p* value)	**7.15 (0.007)**	**1.77 (0.18)**	**12.08 (0.0005)**

Notes: *, statistically significantly different from “much worse/worse” (*p* < 0.05; 2-tailed); ^a^, statistically significantly different from 5th grade (*p* < 0.05; 2-tailed); ^b^, statistically significantly different from 6th grade (*p* < 0.05; 2-tailed); ^c^, statistically significantly different from 7th grade (*p* < 0.05; 2-tailed); ^d^, statistically significantly different from 8th grade (*p* < 0.05; 2-tailed); ^e^, statistically significantly different from 9th grade (*p* < 0.05; 2-tailed); ^f^, statistically significantly different from 10th grade (*p* < 0.05; 2-tailed); ^g^, statistically significantly different from 11th + 12th grades (*p* < 0.05; 2-tailed).

**Table 3 children-09-01346-t003:** Participants’ life satisfaction and somatic and psychological symptoms according to sociodemographic characteristics.

Variables		Life Satisfaction: Median	Somatic Symptoms: Median	Psychological Symptoms: Median
Total group (*n* = 1019)		8.00	16.00	11.00
Gender	Male (*n* = 476), median	9.00	-	-
	Female (*n* = 544)	8.00	-	-
	Mann–Whitney U (*p* value)	**98,073.00 (0.002)**	-	-
Grade	5th (*n* = 121)	8.50	16.00 ^e–g^	12.00 ^e–g^
	6th (*n* = 118)	9.00 ^c–g^	17.00 ^e–g^	13.00 ^e–g^
	7th (*n* = 163)	8.00 ^b^	16.00 ^e–g^	11.00
	8th (*n* = 148)	9.00 ^b^	17.00 ^e–g^	11.00
	9th (*n* = 159)	8.00 ^b^	15.00 ^a–d^	10.00 ^a,b^
	10th (*n* = 134)	8.00 ^b^	15.00 ^a–d^	10.00 ^a,b^
	11th + 12th (*n* = 177)	8.00 ^b^	15.00 ^a–d^	10.00 ^a,b^
	Test statistic (F value)	**1875 (0.002)**	**19.75 (0.002)**	**22.61 (0.0003)**
Change in socioeconomic status	Much worse/worse (*n* = 241)	7 ^†,††^	15 ^†^	9 ^†,††^
	The same (*n* = 557)	8 *^,††^	16 *	12 *
	Better/much better (*n* = 221)	9 *^,†^	16	11 *
	Test statistic (F value)	**48.83 (<0.0001)**	**6.30 (0.04)**	**30.00 (<0.0001)**

Notes: *, statistically significantly different from “much worse/worse” (*p* < 0.05; 2-tailed); ^†^, statistically significantly different from “the same” (*p* < 0.05; 2-tailed); ^††^, statistically significantly different from “better/much better” (*p* < 0.05; 2-tailed); ^a^, statistically significantly different from 5th grade (*p* < 0.05; 2-tailed); ^b^, statistically significantly different from 6th grade (*p* < 0.05; 2-tailed); ^c^, statistically significantly different from 7th grade (*p* < 0.05; 2-tailed); ^d^, statistically significantly different from 8th grade (*p* < 0.05; 2-tailed); ^e^, statistically significantly different from 9th grade (*p* < 0.05; 2-tailed); ^f^, statistically significantly different from 10th grade (*p* < 0.05; 2-tailed); ^g^, statistically significantly different from 11th + 12th grades (*p* < 0.05; 2-tailed).

**Table 4 children-09-01346-t004:** Gender differences related to the remote learning experience.

Remote Learning Variables			Total Group (*n* = 1019): *n* (%)	Gender		Chi-Square (*p*-Value)
				Males (*n* = 476): *n* (%)	Females (*n* = 544): *n* (%)	
Attitudes toward remote learning	Involvement in lessons in the past year	Did not attend lessons	188 (18.4)	89 (18.69)	99 (18.19)	0.26 (0.06)
		Connected and listened to lesson—camera off	427 (41.9)	193 (40.54)	233 (42.83)	
		Connected and not active in lesson—camera on	160 (15.7)	73 (15.33)	87 (15.99)	
		Connected and active in lesson—camera on	245 (24.1)	120 (25.21)	125 (22.97)	
	Feelings toward remote learning	Like very much	132 (13.0)	**76 (15.96)**	**56 (10.29) ***	**11.98 (<0.01)**
		Like slightly	338 (33.2)	140 (29.41)	198 (36.39)	
		Dislike slightly	242 (23.8)	116 (24.36)	126 (23.16)	
		Dislike very much	307 (30.1)	143 (30.04)	163 (29.96)	
	Feelings toward	Like very much	213 (20.9)	102 (21.42)	111 (20.40)	2.72 (0.43)
		Like slightly	291 (28.6)	126 (26.47)	166 (30.51)	
Remote learning and social connectedness	Connection to pedagogical team and school	Very much connected	95 (9.32)	45 (9.45)	50 (9.19)	5.93 (0.20)
		Very connected	296 (29.04)	141 (29.62)	155 (28.49)	
		Slightly connected	357 (35.03)	166 (34.87)	191 (35.11)	
		Slightly disconnected	143 (14.03)	76 (15.96)	67 (12.31)	
		Very much disconnected	129 (12.65)	48 (10.08)	81 (14.88)	
	Connection to peers	Very much connected	291 (28.55)	142 (29.83)	149 (27.38)	**13.69 (<0.001)**
		Very connected	336 (32.97)	180 (37.81)	156 (28.67)	
		Slightly connected	181 (17.76)	75 (15.75)	106 (19.48)	
		Slightly disconnected	116 (11.38)	**40 (8.40)**	**76 (13.97) ***	
		Very much disconnected	96 (9.42)	39 (8.19)	57 (10.47)	

Notes: * Statistically significant difference in percentage between males and females (*p* < 0.05, 2-tailed).

**Table 5 children-09-01346-t005:** Associations of participants’ remote learning experience with sociodemographic characteristics and well-being (*n* = 1019).

Well-Being Measures				Sociodemographic Characteristics		Remote Learning Experience	
Psychological Symptoms	Psychosomatic Symptoms	Life Satisfaction	Self-Rated Health	Change in Socioeconomic Status	Grade		
**0.15**	**0.187**	**0.171**	**−0.13**	0.06	**−0.22**	Involvement in lessons in the past year	Attitudes toward remote learning
**(<0.0001)**	**(<0.001)**	**(<0.0001)**	**(<0.0001**)	−0.06	**(<0.0001)**		
**0.115**	**0.112**	**0.06**	**−0.1**	0.03	**−0.1**	Feelings toward remote learning	
**(<0.0001)**	**−0.0003**	**−0.04**	**(<0.0001)**	−0.07	**(<0.0001)**		
**0.23**	**0.22**	**0.22**	**−0.12**	0.06	**−0.13**	Connectedness with pedagogical team and school	Remote learning and social connected-ness
**(<0.0001)**	**(<0.0001)**	**(<0.0001)**	**(<0.0001)**	−0.06	**(<0.0001)**		
**0.28**	**0.2**	**0.2**	**−0.2**	0.06	**0.11**	Connectedness with peers	
**(<0.0001)**	**(<0.0001)**	**(<0.0001)**	**(<0.0001)**	−0.06	**(<0.0001)**		

Notes: In all well-being variables, except for “self-rated health”, a higher score represents better well-being.

**Table 6 children-09-01346-t006:** Summary of multiple logistic regression analysis for predicting well-being.

Dependent Variable	Predictors	Coefficient	Standard Error	t	*p*-Value	VIF
Self-rated health	Constant	1.49				
	Gender (reference—male)	**−0.11**	**0.05**	**−2.24**	**0.025**	1.01
	Socioeconomic status	**0.08**	**0.02**	**2.99**	**<0.001**	1.00
	Involvement in lessons	**0.05**	**0.02**	**2.15**	**0.031**	1.11
	Feelings toward remote learning	0.01	0.02	0.46	0.640	1.08
	Connectedness to pedagogical team and school	**0.07**	**0.02**	**3.49**	**<0.001**	1.13
	Connectedness to peers	**0.08**	**0.01**	**4.64**	**<0.001**	1.10
	** *Model summary* **	** *F-ratio = 14.03, p < 0.0001, R* ^2^ *adjusted = 0.08.* **				
Somatic symptoms	Constant	17.54				
	Gender (reference—male)	**−1.35**	**0.30**	**−4.48**	**<0.0001**	1.02
	Grade	0.10	0.07	1.43	0.15	1.08
	Socioeconomic status	0.18	0.20	0.87	0.38	1.01
	Involvement in lessons	**0.46**	**0.13**	**3.50**	**0.0005**	1.15
	Feelings toward remote learning	−0.18	0.14	−1.32	0.18	1.13
	Connectedness to pedagogical team and school	**−0.33**	**0.11**	**−3.00**	**0.002**	1.10
	Connectedness to peers	**−0.38**	**0.12**	**−2.99**	**0.0008**	1.14
	** *Model summary* **	** *F-ratio = 10.18, p < 0.0001, R* ^2^ *adjusted = 0.07.* **				
Psychological symptoms	Constant	16.78				
	Gender (reference—male)	**−1.96**	**0.33**	**−5.89**	**<0.0001**	1.02
	Grade	0.04	0.07	0.57	0.56	1.08
	Socioeconomic status	**0.54**	**0.22**	**2.37**	**0.01**	1.01
	Involvement in lessons	**0.45**	**0.14**	**3.10**	**0.002**	1.15
	Connectedness to pedagogical team and school	**−0.70**	**0.14**	**−4.99**	**<0.0001**	1.14
	Connectedness to peers	**−0.66**	**0.12**	**−5.34**	**<0.0001**	1.10
	** *Model summary* **	** *F-ratio = 20.45, p < 0.0001, R* ^2^ *adjusted = 0.14.* **				
Life satisfaction	Constant	8.60				
	Gender (reference—male)	−0.27	0.14	−1.84	0.06	1.02
	Grade	0.006	0.03	0.18	0.85	1.08
	Socioeconomic status	**0.62**	**0.10**	**6.24**	**<0.0001**	1.01
	Involvement in lessons	**0.19**	**0.06**	**3.03**	**0.002**	1.15
	Connectedness to pedagogical team and school	**−0.42**	**0.05**	**−7.77**	**<0.0001**	1.10
	Connectedness to peers	**−0.15**	**0.06**	**−2.47**	**0.01**	1.14
	** *Model summary* **	** *F-ratio = 20.56, p < 0.0001, R* ^2^ *adjusted = 0.14.* **				

Note: For continuous variables, variables that significantly correlated with well-being were included in the model. For categorical variables, only variables that differed in well-being were entered into the model; VIF, variance of inflation factor.

## Data Availability

The author is the sole person who conceived the study, conducted the research, and wrote the article. The data presented in this study are available on request from the author.

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
