# Peer review of "Remote Learning Experience and Adolescents’ Well-Being during the COVID-19 Pandemic: What Does the Future Hold?"

_children, 2022, doi:10.3390/children9091346_

Round 1

Reviewer 1 Report

  • Thank you for the opportunity to read this paper on your research and findings regarding the wellbeing of young people and children in Israel during the remote learning periods of the COVID 19 pandemic. Please consider adding the location of the study to the abstract so that readers can readily identify the sample.
  •  
  • The manuscript is clear, relevant for the field and presented in a well-structured manner. 
  • The cited references are mostly recent publications and relevant. It would be an improvement if there were references to the pre-pandemic measures of wellbeing in the Israeli setting. This would assist readers in drawing relevance to their own settings.
  • The manuscript is scientifically sound and  the experimental design is appropriate to test the hypothesis. The regression analysis makes some interesting reading, and in particular, given coping strategies make an important assertion in your discussion and conclusion, more could be added to the literature review regarding the current research around coping skills of children during the pandemic. The OECD report https://www.oecd.org/coronavirus/policy-responses/combatting-covid-19-s-effect-on-children-2e1f3b2f/ might assist in further considerations around variables. In particular, it seems relevant beyond parental stress, to identify the further health considerations for low SES participants- accomodation, nutrition, etc. It is with some disappointment that the impact of having contracted COVID-19 or not is not a recorded as a somatic variable. If there is a reason for this decision, it would be a good addition to the limitations.
  • The manuscript’s results would be reproducible and the value of the paper would be improved if the survey was included as an appendix.
  • The figures/tables are appropriate however there are some formatting issues that should be addressed before publication. The data are interpreted appropriately and consistently throughout the manuscript, however there are religion categories in the first table and they are not followed up on in the analysis or findings so could be removed. It is possible that these are included to identify social minority participants, and if this is the case, a brief explanation should be added to explain this.
  • Most importantly, there needs to be a clearer indication of the developmental differences across the research participant cohort. There is a lack of clarity around the grouping of ages at various points in the manuscript. Page 10, line three the statement: 

    In all remote learning items, statically significant correlations were found with grade (i.e., with increasing grade, 3 attitudes towards remote learning decreased except for connectedness with peers; Table 4). 

    You proceed to identify in the discussion on page 13 that "children attending elementary schools in Israel had overall more in-person learning opportunities compared to those in higher grades, who continued to learn remotely when the younger children had already returned to school [26], which may explain why their overall remote learning experience was more negative" (lines 61-63). 

    This information gets lost in the discussion and should be prefaced in the introduction to the different age categories. The differing lengths of social isolation is relevant to the interpretation of the findings beyond a point of discussion.

  • The conclusions are consistent with the evidence and arguments presented, with the exception of a more explicit line of argument being drawn through to assist in supporting the asserted need for further explicit teaching of coping strategies. As identified above, this is important to address.
  • There are clear assertions around the importance of child voice as a point of strength and relevance of this study. As such, explanation as to the opportunity to opt in or out of the study by students, and the settings parameters for the administration of the data collection tool should be addressed to meet the standards of adequacy of detail. For this journal in particular, this needs to be addressed.
  •  

Author Response

  • Thank you for the opportunity to read this paper on your research and findings regarding the wellbeing of young people and children in Israel during the remote learning periods of the COVID 19 pandemic. Please consider adding the location of the study to the abstract so that readers can readily identify the sample.

             Thank you, the location has been added to the abstract.

  • The manuscript is clear, relevant for the field and presented in a well-structured manner. 

Thank you.

  • The cited references are mostly recent publications and relevant. It would be an improvement if there were references to the pre-pandemic measures of wellbeing in the Israeli setting. This would assist readers in drawing relevance to their own settings.

We have added information from the following references:

Inchley, J.; Currie, D.; Budisavljevic, S.; Torsheim, T.; Jaastad, A.; Cosma, A.; Kelly, C.; Arnarsson, Á.M.; Samdal, O. Spotlight on adolescent health and well-being. Findings from the 2017/2018 Health Behaviour in School-aged Children (HBSC) survey in Europe and Canada. International report. 2020, 1.

  • The manuscript is scientifically sound and the experimental design is appropriate to test the hypothesis. The regression analysis makes some interesting reading, and in particular, given coping strategies make an important assertion in your discussion and conclusion, more could be added to the literature review regarding the current research around coping skills of children during the pandemic. The OECD report https://www.oecd.org/coronavirus/policy-responses/combatting-covid-19-s-effect-on-children-2e1f3b2f/ might assist in further considerations around variables.

We have added the following information regarding the coping skills of children from the OECD report in the introduction:

“According to a report by the Organization for Economic Cooperation and Development, children’s daily lives have been impacted by the COVID-19 pandemic. Schools, activities, and other leisure organizations for children have been affected, many activities being offered online instead of in-person. Children were expected to spend more time on digital devices in order to stay in school and connected with others.”

https://www.oecd.org/coronavirus/policy-responses/combatting-covid-19-s-effect-on-children-2e1f3b2f/

  • In particular, it seems relevant beyond parental stress, to identify the further health considerations for low SES participants- accommodation, nutrition, etc. It is with some disappointment that the impact of having contracted COVID-19 or not is not a recorded as a somatic variable. If there is a reason for this decision, it would be a good addition to the limitations.

We have added the following to the limitations:

“Another limitation was that we did not include additional variables to the study (e.g., nutrition levels, accommodation), which could have enriched the findings, specifically for low SES participants.”

  • The manuscript’s results would be reproducible and the value of the paper would be improved if the survey was included as an appendix.

Unfortunately, due to ethical considerations and privacy concerns, we are not able to release the survey.

  • The figures/tables are appropriate however there are some formatting issues that should be addressed before publication. The data are interpreted appropriately and consistently throughout the manuscript, however there are religion categories in the first table and they are not followed up on in the analysis or findings so could be removed. It is possible that these are included to identify social minority participants, and if this is the case, a brief explanation should be added to explain this.

We removed the religion categories.

  • Most importantly, there needs to be a clearer indication of the developmental differences across the research participant cohort. There is a lack of clarity around the grouping of ages at various points in the manuscript. Page 10, line three the statement: 

In all remote learning items, statically significant correlations were found with grade (i.e., with increasing grade, 3 attitudes towards remote learning decreased except for connectedness with peers; Table 4). 

You proceed to identify in the discussion on page 13 that "children attending elementary schools in Israel had overall more in-person learning opportunities compared to those in higher grades, who continued to learn remotely when the younger children had already returned to school [26], which may explain why their overall remote learning experience was more negative" (lines 61-63). 

This information gets lost in the discussion and should be prefaced in the introduction to the different age categories. The differing lengths of social isolation is relevant to the interpretation of the findings beyond a point of discussion.

We have added the following information to the introduction:

“Children in different grades also react differently to the different lengths of social isolation.”

  • The conclusions are consistent with the evidence and arguments presented, with the exception of a more explicit line of argument being drawn through to assist in supporting the asserted need for further explicit teaching of coping strategies. As identified above, this is important to address.

We have added the following sentence to the conclusions: “It is imperative that coping strategies are taught to children in order to help them overcome any issues or barriers associated with the COVID-19 pandemic.”

  • There are clear assertions around the importance of child voice as a point of strength and relevance of this study. As such, explanation as to the opportunity to opt in or out of the study by students, and the settings parameters for the administration of the data collection tool should be addressed to meet the standards of adequacy of detail. For this journal in particular, this needs to be addressed.

We have added the following sentence to the Methods: “Students were given the option to participate or not.”

Thank you very much. We believe the reviewers’ comments and our changes greatly improved our paper.

Reviewer 2 Report

Dear author,

The topic and content of the study is relevant and up-to-date, the research is based on a detailed methodology with clearly presented ethical soundness.

The results are interesting and useful for a better organization of the educational process in secondary and high schools.

Nevertheless, several questions and remarks should be mentioned.

In Abstract: is "WB" - well-being?

Introduction is constructed briefly and requires improving:

- in the section: "Risk and Protective Factors for Remote Learning" (page 3) - the gender is declared as a factor, but is not explained, nevertheless, during 2020 many studies were carried out that discovered the gender differences in the coping with remote learning and academical outcome during covid pandemic.

- the description of the research gap should be presented. There are some studies' results, and what is the clear gap? Why the author come back to study the same questions which were previously examined by the mentioned researchers?

Section 2.2. - why the gender structure of the sample was - "69% girls, 31% boys"? Is the structure of the Israeli students aged 11 to 18 so much imbalanced? And the Section 3 (first paragraph of the section and Table 1) another share of females is mentioned - "53.3% of the sample". So, 69 or 53%? 

Section 2.4.:

- if "Questionnaires were distributed to the students between May and July 2021 during school", why the SES change is measured by the period of 30 days? (question: "according to your opinion, in comparison to before corona virus 2019 pandemic, in the past 30 days, what is the socioeconomic status of the family") - e.g., during the Mar-Dec 2020 the SES of the family dramatically dropped down, but, in July 2021 the SES improved - what is the sense of the answer to such a question for the past 30 days?

- Remote learning Experience. The options of answers do not include "I connected to the lesson with camera off and was active" - why? E.g., many students were participating during my lessons without camera, because they had not a camera, but they had got microphone and participating actively in discussions and were asking their questions, giving their questions and making remarks and comments. Why such behavior model was not taken into account in the study presented in the article - why necessarily to have a camera to take active part? 

- Why 3 kinds of scales were chosen - 1-3 points; 1-4 points and 1-5 points for the different questions? In the next section (2.5) the scale 0-10 is also mentioned for the life satisfaction, but another methodology was used ("Life satisfaction was measured by the Cantril’s ladder"). Does the responsibility for the choice of 3-, - or 5- point scale is on the "HBSC methodology protocol"?

Section 4 Discussion.

lines 58-59 - "Female students tended to participate with their camera off compared to male students." - it can be explained with the independent variable of the girls' concerns about their appearance (clothes, make up  of faces, etc.), and this hypothesis could be justified or not, if in the question about connection with camera on or off - it would be not only about camera, but also about the microphone and sound. That is why the conclusion about the gender impact on the experience evaluation ("Male students had slightly more favorable attitudes towards remote learning") is biased by the methodological incorrectness. 

- The younger students are more protected by their parents, that is why their answers about the SES change are more positive or neutral, that from the older students. This reason is not even mentioned among potential influencing factors.

- lines 84-91 - the gender difference in the health and psychological self-rate can be explained with the gender self-evaluation it-self, but also with the social role - during the isolation of families, the females' role in family and in household is much broader, the responsibilities are deeper and wider, traditions require that men work outside and women accomplish householding duties inside homes, and remote work or remote learning influence on the self-rate of female students because they are involved in the tasks of householding due to their physical presence at home.

These remarks and concerns should be answered or, at least, mentioned.

The improved text could be worth publication.

Author Response

The topic and content of the study is relevant and up to date, the research is based on a detailed methodology with clearly presented ethical soundness.

             Thank you

The results are interesting and useful for a better organization of the educational process in secondary and high schools.

             Thank you

Nevertheless, several questions and remarks should be mentioned.

In Abstract: is "WB" - well-being?

Yes, WB meant well-being; we wrote the word out.

Introduction is constructed briefly and requires improving:

- in the section: "Risk and Protective Factors for Remote Learning" (page 3) - the gender is declared as a factor, but is not explained, nevertheless, during 2020 many studies were carried out that discovered the gender differences in the coping with remote learning and academical outcome during covid pandemic.

We have added the following sentence to the “Risk and Protective Factors for Remote Learning” section:

Children in different grades also react differently to the different lengths of social isolation.  previous research has shown that there have been gender differences in the coping with remote learning and academic outcomes in response to digital learning. Stein-Zamir (2020) found that boys may have an advantage over girls regarding online learning due to a higher perceived ability and engagement with computers [3]. However, others have suggested that girls have more negative beliefs about learning via computer and digitally than boys [7,8]. Yet others have suggested that there are no differences between male and female students’ attitudes regarding digital learning [9].

- the description of the research gap should be presented. There are some studies' results, and what is the clear gap? Why the author come back to study the same questions which were previously examined by the mentioned researchers?

We have added this gap to the end of the introduction:

“Though much research has focused on remote learning among children and adolescents, not much has described well-being among this population. Additionally, this is the first study to our knowledge that focuses on the association of remote learning with sociodemographic characteristics and well-being and among children and adolescents in an Israeli setting.”

Section 2.2. - why the gender structure of the sample was - "69% girls, 31% boys"? Is the structure of the Israeli students aged 11 to 18 so much imbalanced? And the Section 3 (first paragraph of the section and Table 1) another share of females is mentioned - "53.3% of the sample". So, 69 or 53%?              

Thank you for pointing out this error. We have changed the mistake to 53.3% girls.

Section 2.4.:

- if "Questionnaires were distributed to the students between May and July 2021 during school", why the SES change is measured by the period of 30 days? (question: "according to your opinion, in comparison to before corona virus 2019 pandemic, in the past 30 days, what is the socioeconomic status of the family") - e.g., during the Mar-Dec 2020 the SES of the family dramatically dropped down, but, in July 2021 the SES improved - what is the sense of the answer to such a question for the past 30 days?

The question pertaining to SES is a validated question derived from the HBSC data. In our country, COVID restrictions were still enacted during data collection (May and July 2021). As we wanted to learn about recent average SES changes, the question asked about the previous month, meaning, April to June, 2021. Changes were compared to before COVID-19 SES.  This information was added to the text (section 2.4).

- Remote learning Experience. The options of answers do not include "I connected to the lesson with camera off and was active" - why? E.g., many students were participating during my lessons without camera, because they had not a camera, but they had got microphone and participating actively in discussions and were asking their questions, giving their questions and making remarks and comments. Why such behavior model was not taken into account in the study presented in the article - why necessarily to have a camera to take active part? 

In our country, all children were required to use their cameras during lessons. Cameras were available to all children.

- Why 3 kinds of scales were chosen - 1-3 points; 1-4 points and 1-5 points for the different questions? In the next section (2.5) the scale 0-10 is also mentioned for the life satisfaction, but another methodology was used ("Life satisfaction was measured by the Cantril’s ladder"). Does the responsibility for the choice of 3-, - or 5- point scale is on the "HBSC methodology protocol"?

Life satisfaction was measured on a scale from 0 to 10. The scale was used as continues measure and not categorical. We reread the description of life satisfaction scale in the outcome measures section and corrected the text.

Section 4 Discussion.

lines 58-59 - "Female students tended to participate with their camera off compared to male students." - it can be explained with the independent variable of the girls' concerns about their appearance (clothes, make up  of faces, etc.), and this hypothesis could be justified or not, if in the question about connection with camera on or off - it would be not only about camera, but also about the microphone and sound. That is why the conclusion about the gender impact on the experience evaluation ("Male students had slightly more favorable attitudes towards remote learning") is biased by the methodological incorrectness. 

This is correct. We added additional interpretation to the gender differences to the discussion. More specifically, we added the following text: However, lower prevalence of females participating with camera off may not be directly related to negative attitudes toward remote learning. It may be related to other factors such as, for example, possible concerns regarding their appearance on camera. Therefore, gender differences in usage of camera should be further explored.

- The younger students are more protected by their parents, that is why their answers about the SES change are more positive or neutral, that from the older students. This reason is not even mentioned among potential influencing factors.

We have added the following sentence to the discussion:

“Younger students are generally more protected by their parents, which could be the reason why their answers regarding the SES change questions were neutral or more positive as compared to the older students.”

- lines 84-91 - the gender difference in the health and psychological self-rate can be explained with the gender self-evaluation it-self, but also with the social role - during the isolation of families, the females' role in family and in household is much broader, the responsibilities are deeper and wider, traditions require that men work outside and women accomplish householding duties inside homes, and remote work or remote learning influence on the self-rate of female students because they are involved in the tasks of householding due to their physical presence at home.

While we agree with the reviewer that working mothers and women generally took on more of the household chores (often in addition to working from home), this was not reflective of the child population. As our study focuses on children, we do not believe this to be the case, and therefore did not add a section about this point.

These remarks and concerns should be answered or, at least, mentioned.

The improved text could be worth publication.

Thank you very much. We believe the reviewers’ comments and our changes greatly improved our paper.

Round 2

Reviewer 1 Report

Thank you for attending to the changes suggested. 

Reviewer 2 Report

Dear authors, 

thank you very much for the detailed reply.

Now, the paper can be accepted, with this remark introduced in the text: this study is not the first about the gender and economic situation on the remote learning attitude and outcome, but, I do not know, if it is the first one about the Israeli students (it is highly likely, that it is).

This research is not "the first study... that focuses on the association of remote learning with sociodemographic characteristics and well-being" - see the article of Feb. 2021, please: https://www.sciencedirect.com/science/article/pii/S0305750X20303521 . It is possible, that this is the first study focused on Israeli setting. 

There are several articles, even, of myself (I also participated in such studies during and after the pandemic), but, of course, not in Israel.

In any case, the paper is worth publishing.

Good luck!